# PAMAM-Calix-Dendrimers: Third Generation Synthesis and Impact of Generation and Macrocyclic Core Conformation on Hemotoxicity and Calf Thymus DNA Binding

**DOI:** 10.3390/pharmaceutics16111379

**Published:** 2024-10-27

**Authors:** Olga Mostovaya, Igor Shiabiev, Daniil Ovchinnikov, Dmitry Pysin, Timur Mukhametzyanov, Alesia Stanavaya, Viktar Abashkin, Dzmitry Shcharbin, Arthur Khannanov, Marianna Kutyreva, Mingwu Shen, Xiangyang Shi, Pavel Padnya, Ivan Stoikov

**Affiliations:** 1A.M. Butlerov Chemical Institute, Kazan Federal University, Kremlevskaya, 18, 420008 Kazan, Russia; olga.mostovaya@mail.ru (O.M.); shiabiev.ig@yandex.ru (I.S.); danya1075@mail.ru (D.O.); pysin_dima@mail.ru (D.P.); timur.mukhametzyanov@kpfu.ru (T.M.); arthann@gmail.com (A.K.); mkutyreva@mail.ru (M.K.); 2Institute of Biophysics and Cell Engineering of NASB, 27 Akademicheskaya St., 220072 Minsk, Belarus; alesiastanovaya@gmail.com (A.S.); viktar.abashkin@gmail.com (V.A.); shcharbin@gmail.com (D.S.); 3State Key Laboratory for Modification of Chemical Fibers and Polymer Materials, Shanghai Engineering Research Center of Nano-Biomaterials and Regenerative Medicine, College of Biological Science and Medical Engineering, Donghua University, Shanghai 201620, China; mwshen@dhu.edu.cn (M.S.); xshi@dhu.edu.cn (X.S.); 4CQM—Centro de Química da Madeira, Universidade da Madeira, Campus Universitário da Penteada, 9020-105 Funchal, Portugal

**Keywords:** dendrimers, PAMAM-calix-dendrimers, DNA, synthesis, self-assembly, toxicity, structure-activity relationships

## Abstract

**Background/Objectives**: Current promising treatments for many diseases are based on the use of therapeutic nucleic acids, including DNA. However, the list of nanocarriers is limited due to their low biocompatibility, high cost, and toxicity. The design of synthetic building blocks for creating universal delivery systems for genetic material is an unsolved problem. In this work, we propose PAMAM dendrimers with rigid thiacalixarene core in various conformations, i.e., **PAMAM-calix-dendrimers**, as a platform for a supramolecular universal constructor for nanomedicine. **Results**: Third generation PAMAM dendrimers with a macrocyclic core in three conformations (*cone*, *partial cone*, and *1,3-alternate*) were synthesized for the first time. The obtained dendrimers were capable of binding and compacting calf thymus DNA, whereby the binding efficiency improved with increasing generation, while the influence of the macrocyclic core was reduced. A dramatic effect of the macrocyclic core conformation on the hemolytic activity of **PAMAM-calix-dendrimers** was observed. Specifically, a notable reduction in hemotoxicity was associated with a decrease in compound amphiphilicity. **Conclusions**: We hope the results will help reduce financial and labor costs in developing new drug delivery systems based on dendrimers.

## 1. Introduction

Interest in dendrimers has not waned since their discovery in the late 20th century [1,2]. These macromolecules differ favorably from traditional synthetic polymers due to their symmetrical and strictly defined structure, consisting of a core, branching units located around it, and terminal groups on a surface of macromolecule [3]. The size of dendrimer molecules in solution is typically less than a few nanometers, making them attractive objects for the design of drug and gene delivery systems. Dissolved dendrimers have a globular shape, while their internal cavities may be capable of binding hydrophobic compounds [4]. A large number of branches provides the presence of multiple terminal groups, which allows the formation of multiple binding centers and the realization of multivalent effect [5]. Thus, the combination of these features makes dendrimers excellent candidates for a wide range of applications, e.g., as catalysts, extraction agents, and nanocarriers [6,7]. A dendrimer functionalized with a sulfur-containing fragment is used in medical practice as the active substance in registered drugs made by Starpharma (Melbourne, Australia) [8,9]. However, along with a number of attractive properties, dendrimers have serious drawbacks, such as high cost and complexity of synthesis, which limit their application [10]. In this regard, an urgent task for increasing the use of dendrimers is to reduce their production cost, which can be achieved by decreasing the number of synthetic steps and using lower generations. In this case, a new challenge arises as the efficiency of interaction of classical dendrimers with different substrates is directly dependent on the generation, i.e., higher generations have more binding centers that can be involved in complexation by both electrostatic (terminal groups) and hydrophobic (internal cavities) interactions. At the moment, a significant barrier to the widespread use of dendrimer-based medical products is also the high hemo- and cytotoxicity of higher generations, which is especially characteristic of cationic poly(amidoamine) (PAMAM) and poly(propylene imine) dendrimers [11,12]. Reduced toxicity is usually achieved by functionalization of the dendrimer surface groups to reduce the positive charge, but this negatively affects their ability to bind to anionic substrates such as DNA and some proteins [13,14,15,16,17]. The importance of studying the interaction with nucleic acids of various compounds cannot be overemphasized. Such interactions are widely used in medical technologies. Binding to DNA, which disrupts its structure and reduces transcription, is used in the treatment of cancer; DNA-based drugs are used to treat a number of non-tumor diseases [18]. DNA binding and compaction has allowed the delivery of genetic material into the cell [19,20,21]. DNA- and RNA-based vaccines have been actively developed in recent years [22,23,24,25,26].

The replacement of a linear dendrimer core with a macrocyclic one proved to be a promising direction for the modification of dendrimers [27,28]. Macrocycles appear to be an excellent platform for the design of dendrimers, providing the desired derivatives with hydrophobic properties and an extensively branched surface by introducing an increased number of dendritic fragments into the molecule. Moreover, macrocycle-based dendrimers have several types of cavities, i.e., along with the cavity formed by dendrons, their structure includes an additional cavity provided by the macrocycle. For example, dendrimers based on β-cyclodextrin had seven dendrons and were capable of binding the hydrophobic antihelminthic drug Albendazole [29]. A dendrimer with a pillar[5]arene core demonstrated the potential for charge transfer in perovskite solar cells [30]. Previously, we found effective binding of drugs and biopolymers (proteins and DNA) by the first and second generation of PAMAM dendrimers with a thiacalix[4]arene core (**PAMAM-calix-dendrimers**) [31,32,33,34]. The efficiency of interaction of **PAMAM-calix-dendrimers** with lysozyme is higher than that of classical PAMAM dendrimers with ethylenediamine core [31]. It is worthy of note that the thiacalixarene is currently a popular platform for the design of compounds with a variety of applications [35,36,37,38]. The attractiveness of the thiacalixarene platform is due to easy functionalization and synthetic availability of various conformational isomers, which allow the binding centers of different natures to be fixed in a strictly defined way in space, as well as the possibility of varying the hydrophilic-hydrophobic balance of the resulting products within wide limits [39]. Successful use of the unique properties of thiacalixarene allows researchers almost unlimited access to new receptors, catalysts, sensors, etc. [40,41,42,43,44]. However, the examples of macrocycle-based dendrimers given in the literature were limited to low generation ones. Thus, most studies have been carried out for the first generation of macrocycle-based dendrimers, while only a few works are devoted to the second generation [32,45].

In summary, we can conclude that the development of universal methods for the synthesis of low and high generations of macrocycle-based dendrimers and discovery of the relationship between their generation and conformation and binding properties is an important research task (Figure 1). This article is devoted to the effect of generation (first, second, and third) and conformation of **PAMAM-calix-dendrimers** on calf thymus DNA binding. Special attention is paid to the relationship between the hemolytic activity of **PAMAM-calix-dendrimers** and their generation and conformation. The obtained data will help reduce the financial and labor costs associated with new drug delivery and receptor systems based on dendrimers.

## 2. Materials and Methods

### General Experimental Information

Detailed information on the equipment, methods, synthesis, and physical–chemical characterization of the compounds studied is presented in the Appendix A.

## 3. Results and Discussion

### 3.1. Synthesis

Previously, we developed an efficient approach to obtain first (**G1**) and second (**G2**) generation **PAMAM-calix-dendrimers** containing the thiacalix[4]arene core in various conformations (*cone*, *partial cone*, and *1,3-alternate*) [32,33]. In this work, we used the described **G1** and **G2 PAMAM-calix-dendrimers** as initial compounds. The previously developed methods were successfully adapted for the synthesis of **G3 PAMAM-calix-dendrimers** (Figure 1). The first step of the synthesis was the reaction of **G2 PAMAM-calix-dendrimers** with methyl acrylate for 48 h at 25 °C (Figure 1). The ninhydrin test (1% ninhydrin solution in ethanol) to determine the presence of primary amino groups of the mixture was negative after 48 h and indicated the absence of terminal amino groups. As a result, **G2.5** half-generations were obtained at a 90–94% yield and were further reacted with excess ethylenediamine. The reaction was carried out for 110 h at 25 °C to ensure complete reaction and complete substitution of all 32 terminal ester groups. Such mild conditions and the use of a large excess of ethylenediamine minimized the probability of side reactions, e.g., incomplete aminolysis, retro-Michael reactions, and intra- and intermolecular cross-linking [46,47,48]. As a result, third generation (**G3**) **PAMAM-calix-dendrimers** containing the macrocyclic core in various conformations (*cone* (**G3-cone**), *partial cone* (**G3-paco**), and *1,3-alternate* (**G3-alt**)) were obtained with an 80–87% yield.

The synthesized **G2.5** and **G3 PAMAM-calix-dendrimers** were fully characterized by physical methods, i.e., ^1^H and ^13^C{^1^H} NMR, FTIR spectroscopy, and high-resolution electrospray ionization mass spectrometry (ESI HRMS) (Appendix A). For comparative evaluation of the spectral characteristics of the first, second, and third generation dendrimers, the ^1^H NMR spectra of **G1-alt**, **G2-alt**, and **G3-alt** are presented in Figure 2. The ^1^H NMR spectra show an increase in the signal intensity of dendritic fragments relative to the signal intensity of the thiacalixarene platform in the series **G1-alt** < **G2-alt** < **G3-alt**, which confirms the increase in the degree of branching of the macromolecule. The broadening of the proton signals of the macrocyclic core and close (*tert*-butyl, aromatic, and oxymethylene) fragments with increasing dendrimer generation is also noteworthy. This is associated with an increase in the overall conformational mobility of the molecule during the growth of the dendritic structure.

IR spectroscopy was also applied to analyze the structures of **G2.5** and **G3 PAMAM-calix-dendrimers**. The IR spectra of the obtained compounds show bands at 1087–1095 cm^–1^, characteristic of arylalkyl esters and corresponding to the dendron linkage to the macrocyclic platform of thiacalix[4]arene (Ar-O-CH_2_). The IR spectrum of **G3** completely lacks the intense band at 1732 cm^–1^, which is present in the IR spectrum of **G2.5** and corresponds to the vibrations of the ester (COOMe) group. This further confirms the complete substitution of all the ester moieties during the synthesis of **G3 PAMAM-calix-dendrimers**. The synthesized dendrimers were additionally characterized by ESI HRMS. An intense peak of polyprotonated molecular ion [M + 16H + Na]^17+^ is observed in the ESI mass spectrum of **G3 PAMAM-calix-dendrimers**. Thus, third generation **PAMAM-calix-dendrimers** containing a macrocyclic core in various conformations (*cone*, *partial cone*, and *1,3-alternate*) were successfully obtained.

### 3.2. Hemolytic Activity and Platelets Aggregation

Under physiological conditions, the outer surface of the erythrocyte membrane, like that of other cells, is negatively charged due to the presence of glycolipids and glycoproteins in its structure. As a result, cationic particles can interact with the membrane, leading to significant changes or damage to the structure and the formation of pores [49,50]. Previously, we found an unusual behavior in **PAMAM-calix-dendrimers** compared to classical PAMAM dendrimers; namely, a decrease in hemolytic activity with increasing generation [32]. However, that study only focused on two generations of dendrimers with a macrocyclic core in *1,3-alternate* conformations. The hemolytic activity of **G1-alt** (20 µM) was 36% (after 3 h) and 61% (after 24 h), while the hemolytic activity of **G2-alt** (10 µM) was less (4% after 3 h and 20% after 24 h) [32]. The study of hemolytic activity of the third generation **PAMAM-calix-dendrimers** confirmed the maintenance of the trend of decreasing toxicity of **PAMAM-calix-dendrimers** with increasing generation, i.e., the hemolytic activity of **G3-alt** (5–50 µM) was less than 5% after 24 h (Figure 3). We also investigated all three stereoisomers of the third (least toxic) generation of **PAMAM-calix-dendrimers** to reveal the influence of the macrocyclic core conformation on the hemolytic activity of the dendrimers (Figure 3). **G3-cone** was found to have the highest hemolytic activity. Typically, *cone* conformation is the most toxic due to its amphiphilic structure (the substituents are located on one side of the macrocyclic platform), and therefore better ability to integrate into RBC bilayer membrane [51]. In the case of **G3-alt**, the hemotoxic effect was insignificant and did not exceed 5% after 24 h. This effect can be explained by the lower density of the cationic charge, which was distributed evenly on the surface leading to a decrease in negative effects during interaction with RBCs.

The mechanisms of platelet aggregation under the action of associates and nanoparticles, which are not agents for the treatment of diseases associated with problems of hematopoiesis and thromboembolism, are generally poorly understood and can differ greatly for different classes of nanomaterials [52,53]. In our work, we also studied the ability of **G3 PAMAM-calix-dendrimers** to induce platelet aggregation. As a result of measuring aggregation in platelet-rich plasma, no differences were found between samples treated with compounds and control samples. Since the studies were carried out in blood plasma, where all coagulation factors are present, it can be concluded that **G3 PAMAM-calix-dendrimers** did not interact with platelets and did not interfere with the work of factors.

### 3.3. Interaction of PAMAM-Calix-Dendrimers with Calf Thymus DNA

We have previously shown the ability of first generation **PAMAM-calix-dendrimers** to interact with salmon sperm DNA [33]. However, the high hemolytic activity of first generation **PAMAM-calix-dendrimers** requires the investigation and comparison of the complexation of different generations (**G2** and **G3**) with DNA. Electron absorption (UV–Vis) spectroscopy was initially used to prove the interaction of the obtained **PAMAM-calix-dendrimers** with calf thymus DNA. To increase the affinity for the polyanionic DNA molecule, **PAMAM-calix-dendrimers** were converted to the hydrochloric ammonium salt form. The UV–Vis spectra of the ammonium salt and neutral forms of first and second generation **PAMAM-calix-dendrimers** have been described in detail previously [31,32,33]. The absorbance of **PAMAM-calix-dendrimers** at 200–315 nm was due to π-π* transitions of the aromatic rings of the macrocyclic fragment and n-π* transitions of the carbonyl groups. The UV–Vis spectra of all three generations were similar, differing only in the value of optical density. The intensity increased significantly in the order **G1** < **G2** < **G3** due to an increase in the number of chromophoric amide fragments (from 12 to 68, respectively), while the number of aromatic rings remained the same. Calf thymus DNA absorbed in the same spectral region, which was due to π-π* electron transitions in the purine and pyrimidine bases of the DNA molecule [54,55]. The UV–Vis spectra of mixtures of DNA with **PAMAM-calix-dendrimers** for all conformations and generations showed a small hyperchromic effect at 260 nm, indicating the interaction of the components (Figure 4 and Appendix A). The rise of the baseline in the long-wave region of the spectrum additionally confirmed the association of components with the formation of associates [56,57].

The complexity of the UV–Vis spectra made it impossible to quantify binding by this method. Another convenient method for studying compound interactions is fluorescence spectroscopy [58]. However, the lack of fluorescent properties in DNA made it necessary to use a fluorescent marker, i.e., ethidium bromide (EB), in this study. EB intercalates into the DNA molecule with a significant increase in emission intensity. We prepared the DNA/EB systems beforehand, and after half an hour of incubation, the dendrimer was added. When the DNA/EB system was irradiated at 525 nm, intense emission with a maximum at 600 nm was observed. The fluorescence intensity decreased significantly in the presence of increasing amounts of **PAMAM-calix-dendrimers** (0.5–60 μM) (Figure 5a, Appendix A). Next, we recorded fluorescence spectra of free EB in the presence of dendrimers (Figure 5b) to verify that the fluorescence quenching was caused by the formation of the ternary DNA/EB/**PAMAM-calix-dendrimers** system and not by the interaction of EB with the dendrimers. The absence of changes in the intensity of the emission spectra in the EB/**PAMAM-calix-dendrimers** system confirmed the nature of the interaction.

The Stern–Volmer plots had a complex and nonlinear shape with several bends, which significantly complicated the calculation of binding constants (Figure 6, Appendix A). The presence of the hyperchromic effect in the electronic absorption spectra indicates the static nature of the quenching due to the formation of complexes (Figure 4 and Appendix A). However, the Stern–Volmer plots clearly indicated several simultaneous processes, both static (with the formation of different types of complexes) and dynamic. Therefore, we decided to qualitatively evaluate the efficiency of the interaction between **PAMAM-calix-dendrimers** and the DNA. For this purpose, we determined **PAMAM-calix-dendrimer** concentrations (C_50_, µM) at which a 50% decrease in the emission intensity, F_50_, of the DNA/EB system from the initial one was observed (F_50_ = (F_0_ − F_min_)/2) (Appendix A). There was a consistent decrease in these concentrations (from 8.44 to 1.10 μM) with increasing generation (Table 1). The differences between the C_50_ of various conformations were maximal for **G1** (3.56–8.44 μM), and was negated for **G2** (3.00–3.22 μM) and **G3** (1.10–1.31 μM). Thus, the influence of the macrocyclic core was highest for the first generation, and **G1-alt** best bound the DNA. This appeared to be due to the lower steric arrangement of the binding ammonium groups in space [59]. With increasing generation, the influence of the macrocyclic platform decreased significantly due to its shielding by substituents, and the C_50_ concentrations became close.

The secondary structure of the biopolymers determines their biological properties. Circular dichroism (CD) spectroscopy is the most sensitive and relatively inexpensive method for establishing this structure. The CD spectrum of the double-stranded calf thymus DNA molecule had a maximum at 275 nm and a minimum at 247 nm, which was characteristic of a mixture of canonical A- and B-forms of DNA [60,61]. The least toxic **G3 PAMAM-calix-dendrimers** were selected to study the effect of dendrimers on the CD spectrum of DNA. The CD spectra of the DNA/**G3** mixtures showed only weak changes with a slight bathochromic shift of the maxima of the CD signals relative to those of the pure DNA solution (Figure 7). Such character of the CD spectra changes indicated non-intercalative binding of **PAMAM-calix-dendrimers** in the grooves of the DNA, since more significant changes could be observed in the CD spectra in the case of intercalation [62,63]. The results obtained correlated well with previously published data for **G1 PAMAM-calix-dendrimers** [33], in the presence of which the preservation of the DNA’s secondary structure was shown.

Significant changes in the CD spectra of the DNA/EB system were observed in the presence of **G3 PAMAM-calix-dendrimers**. In the CD spectrum of the DNA/EB system, the appearance of an additional positive signal with a maximum at 308 nm was observed, in addition to a significant increase in the amplitude intensities of the CD signals of DNA (Figure 8). This band corresponds to the absorption of molecules of the initially nonchiral intercalator, which acquired chirality when incorporated into an optically active biopolymer [64,65]. When **PAMAM-calix-dendrimers** were added to the DNA/EB system, the new band of intercalated EB disappeared. This indicated the loss of chirality of EB due to its removal from the biopolymer molecule. At the same time, the structure of the biomolecule was preserved, as evidenced by the preservation of signals at 245 and 275 nm, characteristic of free DNA. Thus, we can exclude the possibility of loss of chirality of the intercalator due to “unfolding” of the biopolymer molecule. A similar trend was observed for all three conformations of **G3 PAMAM-calix-dendrimers**. This effect can be explained if we take into account the fact that EB is not only intercalated in the DNA molecule, but at the same time its binding is realized mainly along the small groove due to electrostatic interactions with DNA phosphate backbone [66,67]. Previously, thiacalixarenes containing terminal amino groups were found to bind to double-stranded DNA along the major and minor grooves [68]. Thus, we suggest **PAMAM-calix-dendrimers** displaced EB located in the DNA grooves upon binding to DNA, thereby shifting the equilibrium between them and the intercalated dye molecules. This process led to the release of the intercalator from the biopolymer.

Next, the effect of **PAMAM-calix-dendrimer** generation was determined using the least toxic compounds with a macrocylic core in *1,3-alternate* conformation. The CD spectra of the DNA/EB systems in the presence of **G1-alt**, **G2-alt**, and **G3-alt** showed that the maximum decrease of the intercalator signal at 308 nm was observed for **G3-alt**, which may indicate its almost complete removal from the biopolymer molecule (Figure 9). This fact could be easily explained by the increase in the number of ammonium groups binding to the DNA with increasing dendrimer generation.

An important characteristic of compounds is the size of the particles they form in solutions [69,70]. The size of the particles determines both the rate of their deposition and their retention time in the target organs, e.g., at the site of tumor [71]. Dynamic light scattering (DLS) was used to determine the size of the particles formed in solution upon the interaction of DNA with **PAMAM-calix-dendrimers**. All studied **PAMAM-calix-dendrimers** did not form stable self-associates (10 mM Tris-HCl, pH = 7.4) (Appendix A), which correlates well with previously published data for **G1 PAMAM-calix-dendrimers** [33]. However, supramolecular systems DNA/**PAMAM-calix-dendrimers** with monomodal particle distributions were formed when calf thymus DNA (20 µM in nucleotide base pairs) was added to the dendrimers (Appendix A). Only the DNA/**G1-alt** systems were not monodisperse and had visually distinguishable colloidal aggregates in solution (Appendix A). The DNA/**G1-cone** and DNA/**G1-paco** (concentrations of dendrimers equal to 50, 100, and 500 μM) systems were monodisperse with the lowest polydispersity indices (PDI). The smallest particles (160 nm, PDI 0.21) of the DNA/**G1-cone** system were formed at 50 μM of **G1-cone** (Appendix A). The smallest particles (121 nm, PDI 0.19) of the DNA/**G1-paco** system were formed at 500 μM of **G1-paco** (Appendix A).

Further study of the association of **G2 PAMAM-calix-dendrimers** with calf thymus DNA in the same concentration range showed that the DNA/**G2-cone** systems showed the highest PDI in contrast to **G1**-**cone** (Appendix A). In the case of **G2-cone**, the formation of stable systems was observed at 1, 50, and 500 μM, while the PDI was in the range of 0.30–0.40, with a minimum particle diameter of 567 nm. In the case of **G2-paco**, the formation of monodisperse systems (PDI 0.25–0.29) was observed at 1, 50, and 500 μM of **G2-paco**. The smallest particles (148 and 166 nm) for this stereoisomer were recorded at 500 and 50 μM, respectively (Appendix A). The DNA/**G2-alt** systems were least polydisperse (PDI 0.20–0.38). At the same time, the smallest particles (206 nm, PDI 0.20) for this stereoisomer were recorded at 500 μM of **G2-alt** (Appendix A).

For **G3 PAMAM-calix-dendrimers**, the trend was retained (Appendix A). Thus, the least polydisperse systems were DNA/**G3-cone** with dendrimer concentrations of 2 μM (182 nm, PDI 0.16) and 0.16 μM (281 nm, PDI 0.26) (Appendix A). Similar characteristics were observed for the DNA/**G3-paco** systems (2 μM–185 nm, PDI 0.20; 0.16 μM–244 nm, PDI 0.26) (Appendix A). The polydispersity of the DNA/**G3-alt** systems was higher, and the particle size also exceeds that of analogs (from 202 to 532 nm, PDI 0.30–0.42) (Appendix A). Reducing the DNA concentration to 10 μM (in nucleotide base pairs) (Appendix A) resulted in stable monodisperse systems in a wide range of **G3** concentrations. At the same time, the particle sizes decreased significantly. The diameter of the formed particles of the DNA/**G3-cone** system (2 μM of **G3-cone**) was 102 nm (PDI 0.17) (Appendix A). The particles of the DNA/**G3-paco** system were slightly larger (245 nm, PDI 0.26–0.27), especially at low (up to 0.33 μM) dendrimer concentrations. At higher concentrations of **G3-paco**, the particle sizes were 123–185 nm (PDI 0.17–0.23) (Appendix A). The polydispersity of the DNA/**G3-alt** system also decreased (PDI 0.15–0.17) with particle sizes of 106 and 119 nm at 5 and 3.3 μM of **G3-alt**, respectively (Appendix A).

Thus, all three generations of **PAMAM-calix-dendrimers** with calf thymus DNA formed monodisperse systems with submicron (<200 nm) particles. In the case of **G1 PAMAM-calix-dendrimers**, the most significant DNA compaction is achieved for **G1-cone**, which was consistent with the data obtained earlier in the study of the thiacalixarene complexes with calf thymus DNA [72]. Shielding of the macrocyclic core of **G3 PAMAM-calix-dendrimers** by dendrons led to a change in the size characteristics of the formed particles of DNA/**G3**. The particle sizes were found to be close, with the greatest compactification achieved for **G3-alt**, which is the most similar in structure to classical PAMAM dendrimers.

The electrokinetic potentials of DNA complexes (10 μM in nucleotide base pairs) with the third generation **PAMAM-calix-dendrimers** indicated the stability of the as-formed associates (Appendix A). The resulting complexes are generally positively charged (electrokinetic potential ranges from +30.5 to +40.0 mV), with the exception of complexes with very low concentrations of **PAMAM-calix-dendrimers** (1 μM, range from –19.8 to –28.3 mV) (Appendix A). This high positive charge on G3 **PAMAM-calix-dendrimers** without DNA (Appendix A) put them on par with cationic polymers with excellent abilities in transfection [73].

Next, we used the electron microscopy method to evaluate the morphology of the formed particles. We chose DNA-based systems and the least toxic **G3 PAMAM-calix-dendrimers**, which formed particles of minimal size (<200 nm). Figure 10a shows the formation of filamentous structures of a calf thymus DNA solution (0.6 μM in nucleotide base pairs). **G3 PAMAM-calix-dendrimers** formed amorphous particles regardless of core conformation (Figure 10c, Appendix A). For the DNA/**G3-cone** system, monodisperse particles with a diameter of approximately 50 nm are presented in TEM images (Figure 10b), while the particle size of the DNA/**G3-paco** system reached ~100 nm (Appendix A). TEM images of the DNA/**G3-alt** system (Appendix A) show conglomerates of particles composed of smaller particles, which is characteristic of this conformation [74]. Some discrepancy in the sizes of nanoparticles obtained by DLS and TEM was not a surprising fact and could be explained by different methods of sample preparation and study. The DLS method measures the hydrodynamic particle diameter of the colloidal system, while the TEM method measures the size of nanoparticles formed by concentration and subsequent drying on the surface.

## 4. Conclusions

Thus, a simple and efficient method for the synthesis of the third generation **PAMAM-calix-dendrimers** with a macrocyclic core based on *p*-*tert*-butylthiacalix[4]arene has been developed. The replacement of the linear core by a macrocyclic core led to a dramatic decrease in hemolytic activity with increasing generation, which is in contrast to the results for classical PAMAM dendrimers. The conformation of the macrocyclic core also had an effect on the hemolytic activity, which decreased in the series *cone* > *partial cone* > *1,3-alternate*. The macrocyclic core served as a template for the fixation of positively charged ammonium groups. Therefore, the lowest hemotoxicity was observed for *1,3-alternate* stereoisomers possessing the lowest charge density due to the even distribution of substituents on different sides of the macrocycle, while the highest hemotoxicity was found for the most amphiphilic *cone* stereoisomers with maximally separated hydrophilic (positively charged) and hydrophobic parts. Apparently, the influence of the macrocyclic core was due to its hydrophobicity, which allowed hydrophobic interactions to be additionally realized in addition to electrostatic and hydrogen interactions characteristic of classical PAMAM dendrimers. As a result, even the first generation of **PAMAM-calix-dendrimers** was capable of DNA binding. Increasing the generation of **PAMAM-calix-dendrimers** led to an increase in DNA binding efficiency, while the influence of the macrocyclic platform was negated. Thus, the third generation **PAMAM-calix-dendrimers** are analogues of classical PAMAM dendrimers with significantly reduced hemolytic activity. We found that **PAMAM-calix-dendrimers** formed small (d < 200 nm) associates with calf thymus DNA regardless of the generation and conformation of the macrocyclic core. Thus, **G3-alt** with both low hemolytic activity and high DNA binding efficiency was the most promising for the creation of nucleic acid compacting agents among all investigated **PAMAM-calix-dendrimers**. We hope our study will open a wide range of possibilities for the design and synthesis of new macrocyclic dendrimers capable of binding nucleic acids, as well as serve to further expand the range of applications of dendrimer compounds.

## Data Availability

The data presented in this study are available in Appendix A.

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
