# Peer review of "PAMAM-Calix-Dendrimers: Third Generation Synthesis and Impact of Generation and Macrocyclic Core Conformation on Hemotoxicity and Calf Thymus DNA Binding"

_pharmaceutics, 2024, doi:10.3390/pharmaceutics16111379_

Round 1
Reviewer 1 Report
Comments and Suggestions for Authors
This work reports the synthesis of G3.0 PAMAM dendrimers with the macrocyclic core in three conformations. The obtained dendrimers were capable of binding calf thymus DNA.
The findings are interesting, so it can be accepted for publication in this journal after addressing the points below.
The authors refer to the MS spectra but do not directly refer to the parent peak in the paper. In the MS spectrum, we can see peaks with molecular weights larger than [M + 16H + Na]17+. Assign of the highest mass peak of significance in the spectrum is important. Have you ever estimate peaks higher than [M + 16H + Na]17+.
Table S2
Experimental data with different values under the same experimental conditions, G2-cone 50 μM, are listed. Please add some kind of comment in more detail to prevent confusion.
Author Response
Reviewer #1:
This work reports the synthesis of G3.0 PAMAM dendrimers with the macrocyclic core in three conformations. The obtained dendrimers were capable of binding calf thymus DNA.
The findings are interesting, so it can be accepted for publication in this journal after addressing the points below.
Response:
Dear Reviewer! Thank you very much for carefully reading and reviewing our paper! We are very pleased that you have given it such a high rating.
The authors refer to the MS spectra but do not directly refer to the parent peak in the paper. In the MS spectrum, we can see peaks with molecular weights larger than [M + 16H + Na]17+. Assign of the highest mass peak of significance in the spectrum is important. Have you ever estimate peaks higher than [M + 16H + Na]17+.
Response:
The presence of polyprotonated molecular ion peaks is a common pattern in the mass spectra of PAMAM dendrimers [10.1016/S0168-1176(97)00161-4; 10.1007/s00216-012-6673-4].
In our case, the experimental procedure for recording mass spectra involves the use of formic acid (0.1%) as an additive to the eluent. As the studied PAMAM-calix-dendrimers have numerous amino groups in their structure, they are protonated in the presence of acid during mass spectrometry procedure, which does not allow the registration of the parent peak [M]+.
Rarely observed higher peaks correspond to fragmentation products of the parent peak and its protonated forms. The exact assignment of such fragmentation peaks is a complex and time-consuming task, which was not the aim of this study.
Appropriate refinements have been made to the description of the MS experimental procedures:
High-resolution mass spectra with electrospray ionization (ESI HRMS) were obtained on an Agilent iFunnel 6550 Q-TOF LC/MS (Agilent Technologies, Santa Clara, CA, USA) in positive mode, equipped with Agilent 1290 Infinity II LC. Elution solvent used was 0.1% formic acid in mixture of Milli-Q water (20%) and HPLC-grade acetonitrile (80%). ESI conditions: carrier gas-nitrogen, temperature 300 °C, carrier flow rate 0.2 L × min–1, nebulizer pressure 275 kPa, funnel voltage 3500 V, capillary voltage 500 V, total ion current recording mode, 100–3000 m/z mass range.
Table S2
Experimental data with different values under the same experimental conditions, G2-cone 50 μM, are listed. Please add some kind of comment in more detail to prevent confusion.
Response: Unfortunately, there was a typo in Table S2. The required correction has been made, i.e., “50 μM” has been replaced by “500 μM” in the top row of Table S2.

Reviewer 2 Report
Comments and Suggestions for Authors
This is a very important and timely manuscript which should be of high interest to the Pharmaceutics readership. This study describes the importance of changing the core topology of PAMAM dendrimers from linear to macrocyclic (i.e., calix type). This subtle modification led to a significant reduction in hemo-toxicity while also enhancing the binding efficacy of lower generation PAMAM dendrimers (i.e., G=3) with calf thymus DNA.
The manuscript is very well written, well referenced and contains excellent graphics . The authors presented sufficient as well as appropriate experimental details to make all final conclusions by the authors compelling to this Reviewer.
As such, I recommend immediate publication as I offer my sincere congratulations on this very nice work.
Author Response
Reviewer #2:
This is a very important and timely manuscript which should be of high interest to the Pharmaceutics readership. This study describes the importance of changing the core topology of PAMAM dendrimers from linear to macrocyclic (i.e., calix type). This subtle modification led to a significant reduction in hemo-toxicity while also enhancing the binding efficacy of lower generation PAMAM dendrimers (i.e., G=3) with calf thymus DNA.
The manuscript is very well written, well referenced and contains excellent graphics. The authors presented sufficient as well as appropriate experimental details to make all final conclusions by the authors compelling to this Reviewer.
As such, I recommend immediate publication as I offer my sincere congratulations on this very nice work.
Response:
Dear Reviewer! Thank you very much for carefully reading and reviewing our paper! We are very pleased that you have given it such a high rating.

Reviewer 3 Report
Comments and Suggestions for Authors
In this manuscript, the authors developed a method to synthesize 3rd generation dendrimers starting from thiacalixarene cores. Based on different core conformations, the final 3rd generation dendrimers also got different conformations: cone, partial cone, and 1,3-alternate. The authors investigated the effects of conformations on the dendrimer properties and found that the 3rd generation dendrimer originating from the 1,3-alternate core has the least hemolytic toxicity. Other information including polymer characterization and polymer-DNA complexation were also provided to prove that the polymer can successfully bind with DNAs. The content in this manuscript is clear, detailed, and rigorous. Before accepting this manuscript, I recommend the authors answer the following questions:
Comments:
Comment 1: Why was the calf thymus DNA selected in this study? Is it special when compared to other DNAs for example, plasmid DNA? Is this new PAMAM working on RNA binding and delivery?
Comment 2: Zeta potential is needed to evaluate the cationic content of the polymer or polymer/DNA complex.
Comment 3: The agarose gel electrophoresis (AGE) is an easy way to visualize the complexation ability of the polymers at different polymer/DNA (N/P) ratios (either molar or mass ratios). An example of AGE protocol can be found in this literature: Acta biomaterialia 9 (1), 4726-4733
Author Response
Reviewer #3:
In this manuscript, the authors developed a method to synthesize 3rd generation dendrimers starting from thiacalixarene cores. Based on different core conformations, the final 3rd generation dendrimers also got different conformations: cone, partial cone, and 1,3-alternate. The authors investigated the effects of conformations on the dendrimer properties and found that the 3rd generation dendrimer originating from the 1,3-alternate core has the least hemolytic toxicity. Other information including polymer characterization and polymer-DNA complexation were also provided to prove that the polymer can successfully bind with DNAs.
The content in this manuscript is clear, detailed, and rigorous. Before accepting this manuscript, I recommend the authors answer the following questions:
Response:
Dear Reviewer! Thank you very much for carefully reading and reviewing our paper! We are very pleased that you have given it such a high rating.
Comment 1: Why was the calf thymus DNA selected in this study? Is it special when compared to other DNAs for example, plasmid DNA? Is this new PAMAM working on RNA binding and delivery?
Response: Our previous study [10.3390/ijms222111901] described the interaction of the first-generation PAMAM-calix-dendrimers with salmon sperm DNA. In the present study, we chose calf thymus DNA as having a significantly higher molecular mass. Thus, we have strongly proved the compactification of even high molecular weight DNA by PAMAM-calix-dendrimers into small size particles (up to 200 nm). The similar effect of our synthesized compounds on DNAs differing significantly in molecular weight gives us prerequisites for extrapolation of these properties to plasmid DNA as well, but we believe that this requires additional research. The effect of PAMAM-calix-dendrimers on RNA should also be the subject of a separate study, since the difference between RNA and DNA is quite significant. We are likely to observe binding of RNA by PAMAM-calix-dendrimers via electrostatic interactions, but we cannot yet predict this effect based on the present study. Overall, this is a very interesting work, and we plan to expand the range of research objects in the future. Thanks to the Reviewer for valuable advice.
Comment 2: Zeta potential is needed to evaluate the cationic content of the polymer or polymer/DNA complex.
Response: As requested by the Reviewer, we measured zeta potentials. The discussion of the obtained results has been added to the text:
The electrokinetic potentials of DNA complexes (10 μM in nucleotide base pairs) with the third-generation PAMAM-calix-dendrimers indicated the stability of the formed associates (Table S4). The resulting complexes are generally positively charged (electrokinetic potential ranges from +30.5 to +40.0 mV), with the exception of complexes with very low concentrations of PAMAM-calix-dendrimers (1 μM, range from –19.8 to –28.3 mV) (Figures S79–S90). This high positive charge on G3 PAMAM-calix-dendrimers without DNA (Table S4) put them on par with cationic polymers with excellent abilities in transfection [73].
A corresponding experimental section has also been added to the Electronic Supplementary Information
1.9. Electrokinetic Potentials.
Electrokinetic (ζ) potentials were determined by electrophoretic light scattering on the Zetasizer Nano ZS (Malvern Instruments, Worcestershire, UK). Samples were prepared for the DLS measurements and transferred with the syringe to the disposable folded capillary cell for measurement. The ζ potentials were measured using the Malvern M3-PALS method and averaged from five measurements.
Comment 3: The agarose gel electrophoresis (AGE) is an easy way to visualize the complexation ability of the polymers at different polymer/DNA (N/P) ratios (either molar or mass ratios). An example of AGE protocol can be found in this literature: Acta biomaterialia 9 (1), 4726-4733
Response:
In this study, the complexation of PAMAM-calix-dendrimers with DNA was strongly confirmed by UV-Vis, fluorescence and CD spectroscopy. The mechanism of interaction has also been hypothesized and proven. DNA compactization by PAMAM-calix-dendrimers was proven by DLS and TEM. The methods used definitely confirmed complexation and aggregate formation. We fully agree with the Reviewer about the importance of using agarose gel electrophoresis. According to the first recommendation of the reviewer, we plan to further study the interaction of PAMAM-calix-dendrimers with plasmid DNA and RNA using agarose gel electrophoresis. Once again, we thank the Reviewer for recommending the publication (Acta biomaterialia 9 (1), 4726-4733), which helped us to further evaluate the quality of our work. We were pleased to cite it in the manuscript.

Round 2
Reviewer 3 Report
Comments and Suggestions for Authors
The authors addressed my concerns. I recommend accepting this revised version.
Author Response
Reviewer:
The authors addressed my concerns. I recommend accepting this revised version.
Response:
Dear Reviewer! The authors sincerely appreciate the reviewer for their thorough evaluation of the manuscript and the valuable suggestions provided. We are confident that the Reviewer’s recommendations have significantly enhanced the quality of the manuscript.